# Lipolytic Effects of 3-Iodothyronamine (T1AM) and a Novel Thyronamine-Like Analog SG-2 through the AMPK Pathway

**DOI:** 10.3390/ijms20164054

**Published:** 2019-08-20

**Authors:** Michael Rogowski, Lorenza Bellusci, Martina Sabatini, Simona Rapposelli, Shaikh M. Rahman, Grazia Chiellini, Fariba M. Assadi-Porter

**Affiliations:** 1University of Alabama Birmingham School of Medicine, Cardiology Division, 901 19th St. S., Birmingham, AL 35209, USA; 2Department of Pathology, University of Pisa, 56126 Pisa, Italy; 3Department of Pharmacy, University of Pisa, 56126 Pisa, Italy; 4Department of Nutritional Sciences, Texas Tech University, P.O. Box 41270, Lubbock, TX 79409, USA; 5Department of Biochemistry, University of Wisconsin-Madison, 433 Babcock Drive, Madison, WI, 53706-1544, USA; 6Department of Integrative Biology, University of Wisconsin-Madison, 250 N. Mills St, Madison, WI 53706, USA

**Keywords:** 3-iodothyronamine (T1AM), thyroid hormone analogs, lipid metabolism, AMPK pathway, rhodamine (TRITC), cell imaging, metabolic reprogramming, mitochondria

## Abstract

3-Iodothyronamine (T1AM) and its synthetic analog SG-2 are rapidly emerging as promising drivers of cellular metabolic reprogramming. Our recent research indicates that in obese mice a sub-chronic low dose T1AM treatment increased lipolysis, associated with significant weight loss independent of food consumption. The specific cellular mechanism of T1AM’s lipolytic effect and its site of action remains unknown. First, to study the mechanism used by T1AM to gain entry into cells, we synthesized a fluoro-labeled version of T1AM (FL-T1AM) by conjugating it to rhodamine (TRITC) and analyzed its cellular uptake and localization in 3T3-L1 mouse adipocytes. Cell imaging using confocal microscopy revealed a rapid intercellular uptake of FL-T1AM into mitochondria without localization to the lipid droplet or nucleus of mature adipocytes. Treatment of 3T3-L1 adipocytes with T1AM and SG-2 resulted in decreased lipid accumulation, the latter showing a significantly higher potency than T1AM (10 µM vs. 20 µM, respectively). We further examined the effects of T1AM and SG-2 on liver HepG2 cells. A significant decrease in lipid accumulation was observed in HepG2 cells treated with T1AM or SG-2, due to increased lipolytic activity. This was confirmed by accumulation of glycerol in the culture media and through activation of the AMPK/ACC signaling pathways.

## 1. Introduction

3-Iodothyronamine (T1AM) is a recently discovered endogenous hormone like molecule [1,2] structurally and metabolically related to thyroid hormone. Recent studies have demonstrated acute and chronic T1AM treatment to have potent effects on shifting whole body macronutrient metabolism in mammals [3,4]. Haviland et al. demonstrated that chronic low dose administration of T1AM was able to induce significant weight loss in obese mice, and the effect was still maintained three weeks following cessation of the treatment [3]. This weight loss was found to be due to a combination of increased lipolysis, and decreased lipid synthesis while maintaining glucose regulation. These changes in macro nutrient metabolism are robust and present a promising application to aid in the achievement of sustained weight loss in obese humans. 

T1AM was originally discovered as an activator of trace amine-associated receptor 1 (TAAR1, a membrane spanning G-protein coupled receptor). T1AM affects a number of different biological systems, being able to induce dramatic decreases in core temperature, heart rate, and cardiac output in mice [1]. However, the specific mechanism of how T1AM exerts its physiological action is not well understood. Despite TAAR1 receptors being the suspected target of T1AM, it has been demonstrated that TAAR1 receptor knockout mice maintain the wild type response of decreased core temperature when given T1AM [5]. This suggests an alternative mode of T1AM interactions to modulate its properties beyond its ability to interact with TAAR1 receptors. Even less clear are T1AM’s effects at the [6] cellular level specifically. Little is known about its method of entry into the cell and its possible internal targets, which remain unknown. In our previous biodistribution study, carried out by injecting radiolabeled ^125^I- T1AM in the tail vein of mice, we observed that T1AM was still persistent in liver, skeletal muscle, and adipose tissue 24 h after administration, while remaining undetectable in other tissues [7]. These results suggest that T1AM may be acting internally in the major lipid metabolizing tissues. 

The number of in vitro T1AM studies in the literature is scarce [8,9], yet the specific cellular response to T1AM is important to understand how it elicits its whole body effect in vivo. Our previous multidisciplinary in vivo studies provided compelling evidence that the weight loss incurred from T1AM treatment was due to both T1AM’s upregulation of fatty acid oxidation (i.e., increasing 3-hydroxybutyrate, a ketone body) as well as reduction in lipid synthesis [3,6]. To examine this phenomenon more closely, we chose an in vitro adipocyte model, the 3T3-L1 mouse cell line, as a logical platform to gain insight into the cellular mechanisms of T1AM’s ability to affect macronutrient metabolism and contribute to weight loss. Taking into account the non-polar nature of the T1AM molecule itself, and its high-affinity binding to plasma lipoproteins such as apo-B100 [10], we speculated that T1AM might be stored in lipid droplets, where it can modulate cellular signaling pathways to increase lipid oxidation and decrease lipid accumulation. To image T1AM intracellular localization in the 3T3-L1 mouse adipocyte cell line, we synthesized a fluoro-labeled version of T1AM (FL-T1AM) that conjugated T1AM to rhodamine (TRITC) (Figure 1). In addition, since our recent studies have shown that the thyronamine-like analog SG-2 represents a potent mimic of T1AM, with respect to TAAR1 agonist activity as well as metabolic and neurological effects [11,12], we also explored the ability of this compound to modulate lipid metabolism in adipocytes (3T3-L1 cell) at concentrations comparable to our previous in vitro and in vivo studies [3,6,11,13]. Finally, taking into account that liver is the major site of lipolysis and lipogenesis, we also explored the mechanism of T1AM and SG-2 action on both lipid accumulation and lypolysis in cultured human hepatoma (HepG2) cells.

## 2. Results

### 2.1. T1AM Localizes to Mitochondria of Mature 3T3-L1 Cells

Previous studies suggested adipose tissue as a possible storage depot of T1AM after systemic administration [10,13]. To better understand this critical aspect, we used FL-T1AM and confocal microscopy in conjunction with fluorescent labeling of lipids and observed that internalization of T1AM in mature adipocytes is extremely rapid, with FL-T1AM being detected inside cells within seconds of addition to media with no noticeable concentration of FL-T1AM on the cell surface (Figure 2A). Lipid droplet visualization using Nile Red in conjunction with FL-T1AM did not show any overlap in the respective signals (absence of yellow), suggesting that, contrary to our expectation, T1AM does not accumulate in the lipid droplet of adipocytes (Figure 2B,C).

Taking into account T1AM’s structural similarities to thyroid hormone we next explored T1AM distribution in cell nuclei, by assessing FL-T1AM nuclear co-localization with DAPI nuclear stain. Consistent with the results of our previous in vitro investigation in cancer cell lines [14], our present results indicate that T1AM does not enter directly into the nucleus of adipocytes (Figure 2C).

Previous studies have shown that when applied to heart-derived cells T1AM is able to target F_0_F_1_-ATP synthase within mitochondria [15]. Therefore, we next tried to assess whether T1AM may localize in the mitochondria of mature adipocytes. Our results from randomized field images indicate that FL-T1AM is most prominently internalized into the mature adipocytes and co-localized in the mitochondria (Figure 2D).

### 2.2. T1AM Shows a Rapid Cellular Uptake into Adipocytes

During visualization of T1AM uptake using confocal microscopy, our initial observations revealed that FL-T1AM became visible inside the cells within seconds of addition to the media (Figure 2A). To measure the uptake of T1AM in cells we tracked the fluorescent intensity over time using flow cytometry. A uniform uptake pattern of FL-T1AM was observed over the course of multiple repeated experiments. As shown in Figure 3A T1AM uptake appears to involve a three-phase pattern that reaches a plateau after 10–15 min. These three phases consist of a rapid exponential uptake, followed by a linear steady increase, and ending with a plateau. Comparison of fluorescent signal intensities over time at different concentrations (10, 50, 100, and 200–500 nM) are shown in Figure 3B,C, respectively. Even when FL-T1AM was used at the lowest concentration (~10 nM) we were still able to observe a robust and fast response, rapidly reaching a plateau. This observation may suggest a contribution of an active transport mechanism in the uptake of T1AM into cells. 

### 2.3. SG-2 and T1AM Show a Different Efficacy in Decreasing Cellular Lipogenesis

We then sought to determine whether T1AM treatment would have an inverse effect on lipid accumulation in developing 3T3-L1 adipocytes. If so, this would support the weight loss and metabolic shifts observed in our previous study with T1AM treatment in obese wild-type mice [3]. In addition, since the 3-methyldiarylmethane analog of T1AM, namely SG-2, has been recently shown to efficiently reproduce T1AM functional effects in rodents [11,12], we decided to further investigate the potential of this novel compound to modulate lipid accumulation in 3T3-L1 cells. To test whether both T1AM and SG-2 stimulate adipocyte differentiation, we used insulin, dexamethasone, and isobutylmethyl xanthine (differentiation medium, DM) to induce 3T3-L1 preadipocyte differentiations. During the DM induction, varying concentrations of T1AM and SG-2 (1, 2, 5, 10 µM) were added to the medium starting at DAY0 to observe their effects on 3T3-L1 adipocyte differentiation and adipocytes were stained by Oil Red O (ORO) at day 5. Figure 4A shows that SG-2 was able to decrease lipid accumulation in 3T3-L1 cells and plateaued at the highest concentration (10 µM) with very little observable color, while equivalent dosages of T1AM did not result in any significant gross visual changes. Based on the initial visual observations we sought to verify the lipid staining in a more quantitative manner. We used Nile Red lipid staining prior to the imaging experiment to provide a fluorescent signal proportional to the amount of lipid in each cell. Cells were cultured again in the same manner but were instead stained with Nile Red and the fluorescent signal measured in the cells by flow cytometry. Consistent with the visual results, SG-2 treatment during differentiation resulted in a significant reduction in the amount of fluorescent signal detected in mature adipocytes (Figure 4B). In parallel experiments, the dosage of T1AM was increased to 20 µM and its effect was compared to that produced by 10 µM SG-2. The highest concentration (20 µM) of T1AM resulted in a significant reduction in fluorescent signal (Figure 4B). Taken together the findings from Oil Red O staining and flow cytometry confirmed that SG-2 was more potent than T1AM in its ability to prevent lipid accumulation in differentiating adipocytes. 

### 2.4. SG-2 and T1AM Show a Different Efficacy on Lipid Accumulation in Mature Adipocytes

To further investigate the effects of SG-2 and its endogenous parent compound T1AM on lipid accumulation, additional ORO experiments were carried out. Fully differentiated 3T3-L1 adipocytes (DAY8) were incubated with increasing doses (1–50 µM) of both compounds. As shown in Figure 5, T1AM and SG-2 significantly reduced lipid accumulation in adipocytes, and again with SG-2 showing a higher potency than T1AM.

### 2.5. T1AM and SG-2 Induce Lipolysis in HepG2 Cells

Abnormal triglyceride accumulation in the form of lipid droplets can occur in hepatocytes of obese subjects. In addition, dramatic lipid accumulation was previously observed in HepG2 cells that are were treated with steatosis-inducing compounds such as chloroquine [15]. Conversely, triglycerides stored in these lipid droplets can be hydrolyzed into free fatty acids and glycerol that are subsequently released into the surrounding cell culture medium. The amount of glycerol released into the medium is proportional to the complete hydrolysis of triglycerides to free fatty acids. 

The Oil Red O staining was first used as a qualitative measure of total lipid accumulation in cells. Results from the ORO staining are shown in Figure 6A. HepG2 red lipid droplets show a significant (*p* < 0.05) dose-dependent decreasing trend after treatment with T1AM and SG-2. Measurements of glycerol release in media are shown in Figure 6B after incubation with test compounds. Consistent with depleted lipids in cells, results elicit marked glycerol release into media in a dose-dependent manner after exposure to both T1AM and SG2. These results indicate that these compounds are also effective in inducing lipolysis in HepG2 cells. 

### 2.6. Effects of T1AM and SG-2 on HepG2 Cell Viability

Treatment with T1AM or SG-2 at 10 and 25 µM did not significantly cause cell death with respect to control cells (Figure 6C), in agreement with the results from our previous investigations that showed cell viability only started to decrease at 60 µM for SG2 and 125 µM for T1AM in HepG2 cells [12].

### 2.7. Effects of T1AM and SG-2 on AMPK Activation

AMPK is a key player in regulating energy balance at both cellular and whole-body levels, placing it as an ideal therapeutic target for the treatment of altered energy metabolism, which occur in conditions like insulin resistance, type 2 diabetes, and the metabolic syndrome. Once activated, AMPK phosphorylates and inactivates a number of metabolic enzymes involved in lipid metabolism in order to maintain liver energy status. Phosphorylation by AMPK of acetyl-CoA carboxylase (ACC), which is an important rate-limiting enzyme in fatty acid biosynthesis, leads to a fall in malonyl-CoA content and a subsequent decrease in fatty acid synthesis concomitantly with an increase in β-oxidation [16,17,18].

To investigate whether T1AM and its newly developed analog SG-2 regulate metabolism in HepG2 cells via the AMPK pathway, the protein levels of phosphorylated AMPK (p-AMPK), and phosphorylated ACC (p-ACC) were examined. As demonstrated in Figure 6D, 24 h of treatment with 25 µM T1AM or SG-2, significantly (*p* < 0.05) increased the expression of p-AMPK. Notably, concomitantly to AMPK stimulation a significant ACC phosphorylation was also observed (Figure 6E).

## 3. Discussion

T1AM was originally identified as a ligand for the trace amine associated receptor type 1 (TAAR-1), a recently identified new family of GPCR localized mostly on the cell surface, and not necessarily thought to be internalized. Although it is not known if the internalization of T1AM is necessary for exerting its physiological effects, recent studies suggested that treatment with pharmacological doses of T1AM results in internalization into cancer cells or accumulation in tissues [7,14]. Our imaging data using FL-T1AM clearly shows a rapid uptake and internalization of T1AM into 3T3-L1 mouse adipocytes, where it specifically co-localizes to mitochondria. This rapid uptake of T1AM suggests an active transport mechanism. Although we could not specifically identify T1AM transporters in this study, on the basis of T1AM’s structural relationship to thyroid hormone, we can speculate that members of the wide family of thyroid hormone transporters, including sodium taurocholate co-transporting polypeptide, L-type transporters, organic anion transporting polypeptides, and monocarboxylate transporters [19] might play a role in T1AM’s mechanism of cellular uptake.

Interestingly, T1AM does not appear to accumulate in the lipid droplets of adipocytes. This helps to potentially rule out lipid droplets as possible storage depots of T1AM, which we had initially hypothesized on the basis of a previous study demonstrating long-term persistence of radio-labeled T1AM in adipose tissue [7]. However, these preliminary results may require additional verification with much longer incubation times to rule out the possibility that T1AM is not ultimately stored within cellular lipids over a longer time course. In this study, we investigated the effects of SG-2 and T1AM on total lipid accumulation by using the lipid Oil Red O (ORO) staining kit, which is suitable for selective staining and detection of neutral lipids, including triglycerides and cholesterol esters, in cultured 3T3-L1 cells [20]. However, from using this assay alone we could not precisely assess whether the decreased lipid accumulation observed after treatment with T1AM and/or SG-2 was correlated with levels of total cholesterol or accumulation of cholesterol esters vs. triglycerides (TG) only. In the future, we recommend to measure cholesterol content by using a specific staining kit, such as the Amplex Red Cholesterol Assay Kit, as reported by Mitrofanova et al. [21]. In regard to our previous in vivo study [6], we measured total serum cholesterol and TG. Our findings showed total serum cholesterol level decreased while total serum TG increased at 25 mg/Kg T1AM, indicating that the pool of these lipid species are from major tissues including adipose and liver and is due to the T1AM treatment [6]. 

Our previous research utilizing cancer cell lines demonstrated that T1AM is localized to the mitochondria of the cells, which may be important to T1AM and SG-2′s ability to alter macro nutrient utilization from inside the cell. This is consistent with the findings of Cumero et al. [22], who demonstrated that T1AM binds with F_0_F_1_-ATP synthase utilizing mitochondria isolated from bovine heart tissue. Low nanomolar concentrations were able to increase ADP-stimulated mitochondrial respiration, while higher concentrations resulted in inhibition of the enzyme, which was mediated by multiple binding sites of differing affinities. These insights are consistent with our current observations of decreased lipid accumulation at pharmacological treatment doses in adipocytes, along with our previously observed decreased viability measurements in cancer cells at much higher concentrations [14]. 

Due to its ability to reduce weight and increase lipid metabolism in in vivo models, we expected reduced lipid accumulation in adipocytes after treatment with T1AM. Our initial experiments revealed that T1AM treatment on 3T3-L1 cells over the course of differentiation to adipocytes impacts lipogenesis only when used at a 20 µM dose. Notably, our recently developed T1AM analog SG-2, showed a significant activity even when used at 10 µM, thus, suggesting it is considerably more potent than its natural counterpart. Comparable results were also obtained after treatment on 3T3-L1 cells completely differentiated to mature adipocytes. Both compounds appeared to induce a significant lipolytic effect, with SG-2 again showing higher potency as compared to T1AM. For a more quantitative analysis of lipid accumulation, adipocytes treated with SG-2 or T1AM were tagged with Nile Red fluorescent lipid stain to assess the degree of lipid accumulation via flow cytometry (Figure 4B). The flow cytometry data corroborate the visual observations gained from the Oil Red O staining. Indeed, SG-2 treatment was found to be more effective at decreasing the fluorescent signal from Nile Red lipid staining, as compared to T1AM when used at the same dosage. 

Our results showed that T1AM and SG-2 similarly reduced total lipid accumulation in HepG2 cells grown under lipogenic conditions (Figure 6). The observed increase in glycerol release and phosphorylation of AMPK and its downstream lipid regulator, ACC, provide strong evidence that this reduction in lipid accumulation is likely due to the ability of T1AM and SG-2 to induce lipolysis in these cells (Figure 6). Although genetic evidence for a lipolytic pattern after T1AM administration in vivo has been recently provided [8], the molecular mechanism by which T1AM affects lipid metabolism has not been completely clarified yet. Our results suggest that by activation of the AMPK/ACC signaling pathway T1AM might increase lipid breakdown. This is consistent with AMPK’s physiological role in regulating cellular energy metabolism by sparing glucose as cellular energy state diminishes [23]. By demonstrating reduction in lipid accumulation in both adipose and liver tissue cell cultures and identifying a candidate cellular pathway, our in vitro experiments confirm and help to better characterize T1AM’s observed anti-obesogenic properties seen in previous in vivo studies [3,4]. Additionally, in reference to our previous work [12], the concentration at which T1AM and SG-2 induce a reduction in liver lipid accumulation are well below the levels exhibiting cytotoxicity (Figure 6), suggesting a promising potential as therapeutic agents for lipid metabolism. 

## 4. Materials and Methods

### 4.1. Development of Fluorescently Labeled T1AM (FL-T1AM)

The specific mechanism of T1AM entry into the cell, as well as its internal target or areas of compartmentalization, remains unknown. To address these aspects, a fluoro-labeled version of T1AM (FL-T1AM) was synthesized by conjugating T1AM to rhodamine, previously shown to be effective in creating fluoro-conjugates of thyroid hormone derivatives [23,24], and further utilized in our lab to successfully image T1AM in cancer cells [14]. Briefly, rhodamine-labeled T1AM was obtained by combining a 1:1 molar ratio of TRITC and T1AM in 0.1 mL pyridine (Pyr):water (H_2_O):triethylamine (NEt3) (9:1.5:0.1, *v*/*v*/*v*) mixture with stirring for 4 h at room temperature in the dark (Figure 1). The crude product was first purified by flash column chromatography (Hexanes/IPA 2:1) in dark conditions to separate the conjugated and unconjugated forms of T1AM. The desired product was then eluted by 100% methanol wash. Methanol was dried and the dried sample was kept at −20 °C. The powdered residue was reconstituted into a working dilution with DMSO prior to its use.

### 4.2. Cell’s Culture and Treatment

4.2.1. 3T3-L1 Mouse Pre-Adipocyte Differentiation and Flow Cytometric Determination of Lipid Content by Staining with NILE Red 

The 3T3-L1 mouse pre-adipocyte cell line, obtained from Sigma-Aldrich (St. Louis, MO, USA), was selected to provide a model of lipid metabolism in a lipid storage tissue apart from the influence of the whole body physiology in vivo, that would also provide a basis of comparison to the results seen in our previous in vivo mouse study [3]. Prior to differentiation the pre-adipocyte cells were cultured in standard growth media (high glucose DMEM with pyruvate and L-glutamate, supplemented with 10% fetal bovine serum, 1% Penn/Strep) until reaching ~90% confluence. Cells were differentiated upon reaching desired confluency with the addition of 0.5 mM isobutylmethylxanthene (IBMX), 2 µg/mL insulin, and 0.25 µM dexamethasone to the growth medium (differentiation medium, DM) for two days (day 0–2). After two days media were switched to growth media supplemented with insulin only for two days (day 2–4), after which cells were given standard growth media for 4–6 days (day 4–8) until differentiation was complete as evidenced by lipid filling in fully differentiated adipocytes. In order to quantitatively assess lipid accumulation in cultured adipocytes, flow cytometry measurements were conducted using Nile Red fluorescent lipid stain (Sigma Aldrich, St. Louis, MO, USA) with an excitation/emission of 636/552 nm. Cells were incubated in serum free media at 1mg/mL of Nile red staining for approximately 20 min. Cells were then lifted from the plate and placed on ice until ready to be measured at a concentration of approximately 1 million cells in 1 mL. Channel gating was restricted to exclude cellular debris and cell doublets. Each data acquisition was 10,000 cells over 30 s with 1 min between readings over the course of 15–20 min depending upon uptake plateau and available cells per sample with 6 measurements conducted per treatment group.

#### 4.2.2. Intracellular Localization of FL-T1AM

Differentiated 3T3-L1 cells were plated on 35mm glass bottom confocal imaging plates (MatTek, Ashland, MA, USA). Prior to imaging with the fluorescent labeled T1AM (FL-T1AM) cells were incubated with a combination of dyes for 20–40 min in serum free media, and then washed with serum free media three times prior to imaging to minimize background signal. The combination of cells dyes used in each experiment varied depending on the specific cell regions imaged, and were selected to avoid spectral overlap: CellMask™ Green Plasma Membrane Stain, 158 excitation/emission 522/535 nm at 5 ng/mL (Invitrogen, Waltham, MA, USA), MitoTracker^®^ Deep Red, 159 FM excitation/emission ~644/665 nm at 150 nM (Invitrogen, Waltham, MA, USA), and DAPI nuclear stain, 160 excitation/emission 358/461 at 1 μg/mL (Invitrogen, Waltham, MA, USA). DAPI nuclear stain excitation/emission 475/405 at 1 µg/mL (Invitrogen, Waltham, MA, USA), and Nile Red excitation/emission 636/552 1 mg/mL. Confocal images were taken using the Olympus IX81 scope (Olympus Corporation, Tokyo, Japan) with a Yokagawa X1 Spinning Disk confocal box. Laser launch contains four lasers, 405 nm, 488 nm, 561 nm, and 647 nm. The images were captured at 60× using a Photometrics Evolve 512 EMCCD camera and images were analyzed in Slidebook (3i Intelligent Imaging Innovations, Denver, CO, USA). Once baseline cell images were taken FL-T1AM, excitation/emission 561/610 nm, was added to cells at 10 nM concentration and imaged over time (~700 ms exposure per filter, three filters per image) to assess T1AM uptake and cellular localization. 

#### 4.2.3. FL-T1AM Cellular Uptake by Flow Cytometry

To assess the rate and pattern of T1AM uptake into the 3T3-L1 cells we utilized our FL-T1AM compound as a tracer to track its emission signal in cells over time using flow cytometry. Measurements were conducted on a BD fluorescent activated cell sorting Aria III (BDbioscience, San Jose, CA, USA) with excitation/emission detection at 561/610 nm; channel gating was restricted to exclude cellular debris and cell doublets. Each data acquisition was 10,000 cells over 30 s with 1 min between readings over the course of 15–20 min depending upon uptake plateau and available cells per sample. Approximately 1 million cells in 1 mL kept on ice were sampled prior to the addition of FL-T1AM to establish baseline readings. FL-T1AM was added at 100 nM and immediately measured for emission signal detection then measured repeatedly to assess shift in signal detection over time at room temperature. Data are presented as average fluorescence signal based on three repeated measurements at 100 nM. Additionally, differences in uptake across concentrations were measured at a series of concentrations (10, 50, 100, 200, and 500 nM) to evaluate whether signal detection displayed a dose response relationship.

#### 4.2.4. Oil Red O (ORO) Staining of Lipid Accumulation in 3T3-L1 Adipocytes after Treatment with Test Compounds.

3T3-L1 pre-adipocytes were treated with varying concentrations of T1AM or SG-2 (1, 2, 5, 10 µM) at the beginning of the differentiation process to adipocytes (day 0, see above) to assess the impact of treatment on lipid accumulation. At the end of the differentiation period (day 5), adipocytes were fixed in phosphate buffered saline (PBS) with 4% paraformaldehyde for 30–40 min at room temperature. The fixative was aspirated and the cells were rinsed three times with PBS. After fixation and washing of the cells Oil Red O (ORO) solution was added to cover the wells. The ORO stock solution was prepared by dissolving 0.35 g of Oil Red O (Sigma Aldrich, Milan, Italy) in 100 mL of isopropanol by gentle heating and then cooled and filtered through a 0.45 µm filter. The working solution was prepared by diluting three parts of the stock solution in two parts of MilliQ water (stock solution: MilliQ water; 3:2 *v*/*v*). The cell images were captured with a Leitz Fluovert FU (Leica Microsystems, Wetzlar, Germany) microscope. Lipids appeared red. For quantitative analysis of cellular lipids, 1 mL isopropanol was added to each well of the stained culture plate. The extracted dye was immediately removed by gentle pipetting and its absorbance was measured at 510 nm. 

In another set of experiments fully differentiated adipocytes (day 8) were exposed for 24 h to the treatment with increasing doses of T1AM or SG-2 (1, 10, 25, and 50 μM). A total of 10 μM isoproterenol (ISO) was used as positive control. After fixation and washing of the cells, ORO staining was performed as previously described.

#### 4.2.5. Human Hepatocellular Carcinoma (HepG2) Cell Culture and Treatment with T1AM and SG2

Human hepatocellular carcinoma cells (HepG2), obtained from American Type Culture Collection (Manassas, VA, USA), were cultured in low-glucose (LG) Dulbecco’s modified Eagle’s medium (DMEM) (5.5 nM, Invitrogen, Carlsbad, CA, USA) or high-glucose (HG) DMEM (30 nM) supplemented with 10% fetal bovine serum (FBS) and 10 mg/mL penicillin/streptomycin in an atmosphere of 5% CO_2_ at 37 °C. These cells were then treated with 10 and 25 μM concentrations of test compounds for 24 h. After treatments, cells were lysed in a buffer containing 20 mM Tris–HCl (pH 7.5), 0.9% NaCl, 0.2% Triton X-100, and 1% of the protease inhibitor cocktail (Sigma-Aldrich, Milan, Italy) and then stored at −80 °C for further western blot analysis. All the analyses were conducted on cells between the third and the sixth passage.

#### 4.2.6. Western Blotting Analysis

Proteins (20–30 μg) were separated on CriterionTGX TM gel (4–20%) and transferred on Immuno-PVDF membrane (Biorad, Milan, Italy) for 30 min. Blots were incubated with the primary antibody at 4 °C overnight. All the polyclonal antibodies for the protein of interest were purchased from Santa Cruz Biotechnology, Inc., Dallas, TX, USA. Then, blots were washed three times for 10 min with 1X TBS, 0.1% Tween^®^ 20 and incubated for 2h with secondary antibody (peroxidase-coupled anti rabbit in 1X TBS, 0.1% Tween^®^20). After washing three times for 10 min, the reactive signals were revealed by an enhanced ECL Western Blotting analysis system (Amersham, Milan, Italy). Band densitometric analysis was performed using Image Lab Software (Biorad, Milan, Italy). 

#### 4.2.7. Oil Red O (ORO) Staining of Lipid Accumulation in HepG2 Cells

Total lipid accumulation was evaluated according to the method previously described by Liu et al. [25]. Cells were seeded at a density of 3.5 × 10^4^ cells/well in 1 mL of lipogenic growth medium (HG-DMEM) and treated for 24 h with compound T1AM or SG-2, at two different doses, 10 and 25 μM. Cloroquine (25 μM) was used as positive control. Subsequently, after collecting the growth media to be used to perform glycerol level measurements (as detailed below), cells were rinsed twice with PBS and fixed in 4% paraformaldehyde in PBS at 4 °C for 30 min. After three washes with cold PBS, cells were stained with ORO working solution, prepared as described above, for 30 min at room temperature and subsequently rinsed again with PBS. 

#### 4.2.8. Determination of Glycerol Release from HepG2 Cells

After treatment with test compounds, cell culture supernatants were collected from each well and placed in glycerol-free containers. A 125 μg/mL glycerol standard solution (Abcam, Milan, Italy) was used to make a #1 through #8 standard curve. Twenty-five microliters of each supernatant and standard were then transferred into a 96-well plate and 100 μL of either Free Glycerol Assay Reagent (Abcam, Milan, Italy) or MilliQ water added to each well. After incubation at RT for 15 min, glycerol levels were measured by reading absorbance at 540 nm (Bio-Rad 680, Milan, Italy).

#### 4.2.9. Cell Viability by MTT Assay

HepG2 cells were seeded in a 96-well plate at a density of 1.0 × 10^4^ cells/well with DMEM (200 μL/well), and then incubated for 24 h according to routine procedure. After being treated with test compounds (10–25 μM) and incubated for 24 h (eight wells for each sample), 10 μL/well MTT (5 g/L) was added to each well. The medium was then removed after 4 h incubation and 100 μL/well sodium dodecyl sulfate (SDS)-HCl solution was added to dissolve the reduced formazan product after incubation for 16 h at 37 °C and 5% CO_2_. Finally, after mixing each sample using a pipette, the plate was read at 570 nm with a micro-plate reader (Bio-Rad 680, Milan, Italy).

#### 4.2.10. Statistical Analysis

Statistical analyses were performed using GraphPad Prism version 6.0 for Windows (GraphPad Software, San Diego, CA, USA). Data were subjected to one-way analysis of variance for mean comparison, and significant differences among different treatments were calculated according to Tukey’s HSD (honest significant difference) multiple range test. Data are reported as mean ± SEM. Differences at *p* < 0.05 were considered statistically significant. All the analyses were performed in triplicate

## 5. Conclusions

Contrary to our initial assumptions about the functional uptake of T1AM, our imaging data show that rather than primarily interacting with cell surface receptors or by passive diffusion across the cell membrane, T1AM is conceivably internalized into the cell through an active transport mechanism. This provides the basis for future studies to explore the specific transporters responsible for this rapid uptake. Furthermore, we found no evidence of T1AM localizing into the lipid droplet, suggesting that despite its nonpolar nature, lipid storage depots are not also storage sites for T1AM in the cell. This indicates that adipose tissue may not function as the storage of endogenous T1AM in the body. Our in vitro experiments in both 3T3-L1 and HepG2 cells suggest T1AM actively increases cellular lipolysis and that its ability to modulate lipid metabolism may be mediated, at least in liver, through AMPK/ACC pathway activation. Moreover, the synthetic T1AM analog, SG-2, proved quite effective in its ability to prevent lipid accumulation in 3T3-L1 adipocytes and HepG2 hepatocytes, supporting the need for future in vivo studies that may serve as a development and refinement of the anti-obesity applications initially observed with T1AM.

## Figures and Tables

**Figure 1 ijms-20-04054-f001:**
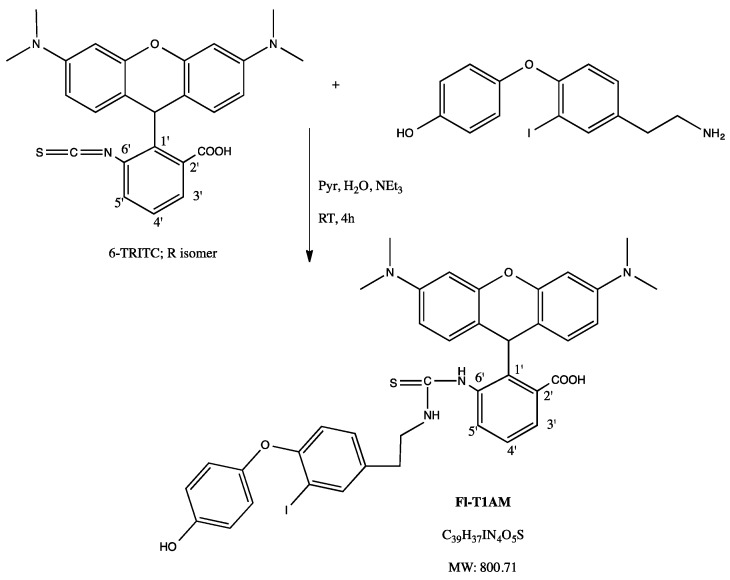
Synthesis of fluorescently labeled T1AM (FL-T1AM). One-step conjugation of Tetramethylrhodamine-6-isothiocyanate (6-TRITC) with T1AM efficiently produces fluorescently-labeled T1AM (FL-T1AM).

**Figure 2 ijms-20-04054-f002:**
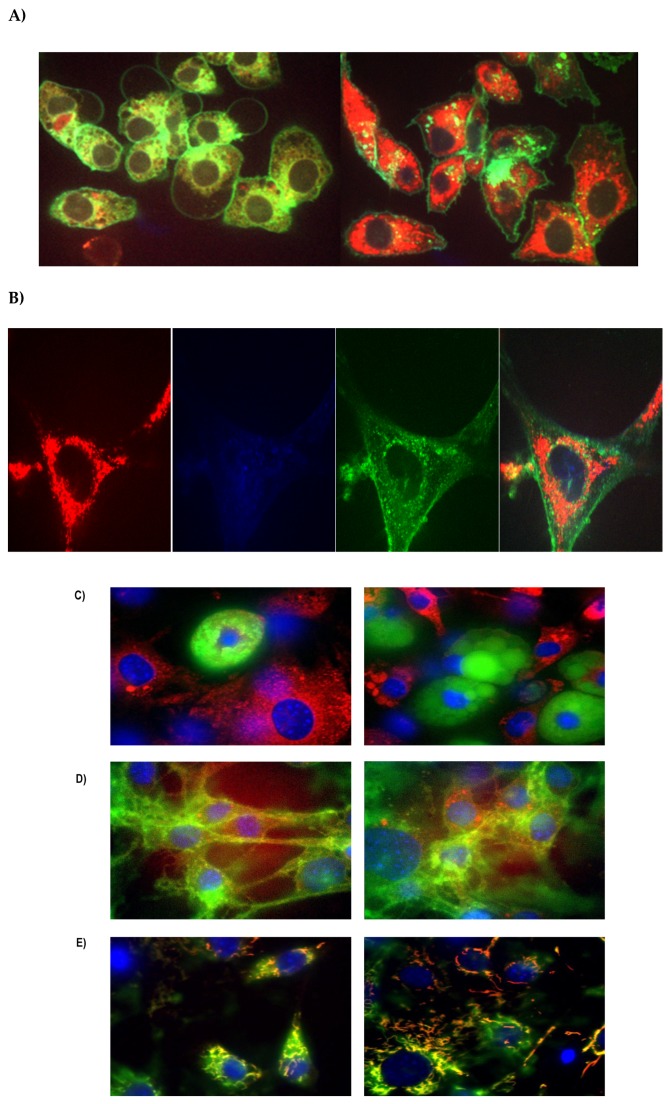
Fluorescent labeling of internalized T1AM in differentiated 3T3-L1 cells. (**A**) Internalization of T1AM is extremely rapid, with FL-T1AM being detected inside cells within seconds of addition to media. (**B**) Snapshots of a zoomed cell to show each individual staining (i.e., Green: Cell membrane stain, Blue: DAPI nuclear stain, Red: FL-T1AM), and composite picture of stains for ease of visualization. (**C**) FL-T1AM does not co-localize into the lipid droplet of mature adipocytes (i.e., lack of yellow signal resulting from red and green overlap). Blue: DAPI nuclear stain, Green: Lipid stain, Red: FL-T1AM. (**D**) FL-T1AM remains perinuclear, but not within the nucleus when internalized into adipocytes. Blue: DAPI nuclear stain, Green: Cell membrane stain, Red: FL-T1AM. (**E**) FL-T1AM co-localizes to mitochondria in 3T3-L1 adipocytes. Blue: DAPI nuclear stain, Green: Mitochondria membrane stain, Red: FL-T1AM.

**Figure 3 ijms-20-04054-f003:**
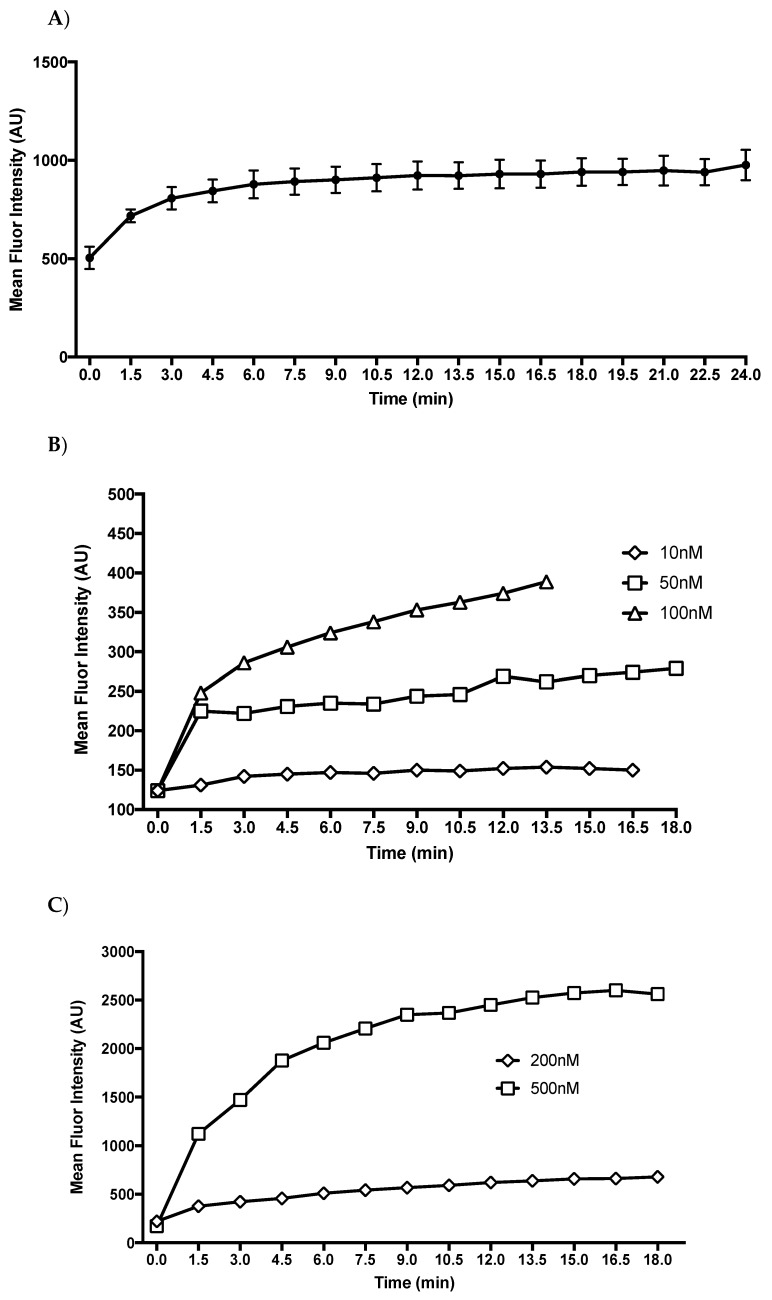
Uptake rate of fluorescent labeled T1AM (FL-T1AM). 3T3-L1 cells were given FL-T1AM and their mean shift in fluorescent intensity was measured with 10,000 cells per time point. T1AM uptake appears to involve a three-phase pattern, consisting of a rapid exponential uptake, followed by a linear steady increase, and ending with a plateau. (**A**) Mean fluorescent shift of FL-T1AM signal at 100 nM ± SEM. (**B**) Fluorescent signal intensity over time at 10, 50, and 100 nM. (**C**) Fluorescent signal intensity over time at 200 nM and 500 nM.

**Figure 4 ijms-20-04054-f004:**
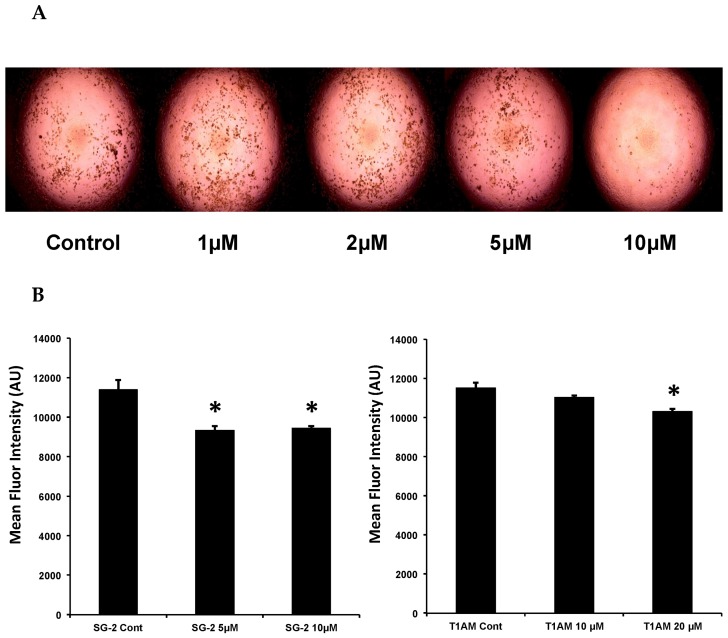
T1AM and SG-2 treatments are associated with decreased lipid accumulation. 3T3-L1 adipocytes were treated with T1AM or SG-2 (1, 2, 5, 10 µM) at the beginning of differentiation (day 0) (**A**) Visible decreases in lipid accumulation were readily observable at day 5 with SG-2 treatment. Treatment with T1AM did not result in visibly detectible decreases in lipid accumulation (pictures not shown). (**B**) Fluorescent intensity of Nile Red lipid stain in 3T3-L1 differentiated adipocytes treated with 20 µM T1AM and 10 µM SG-2 and measured by flow cytometry. Nile red is a fluorescent stain that integrates into and fluoresces when in contact with lipid, allowing for measured fluorescent intensity to correspond to the degree of lipid present in cells. Flow cytometry data are presented as mean channel fluorescence, with +/− SEM. * denotes a significant difference from control at *p* < 0.05 (ANOVA Dunnett’s test).

**Figure 5 ijms-20-04054-f005:**
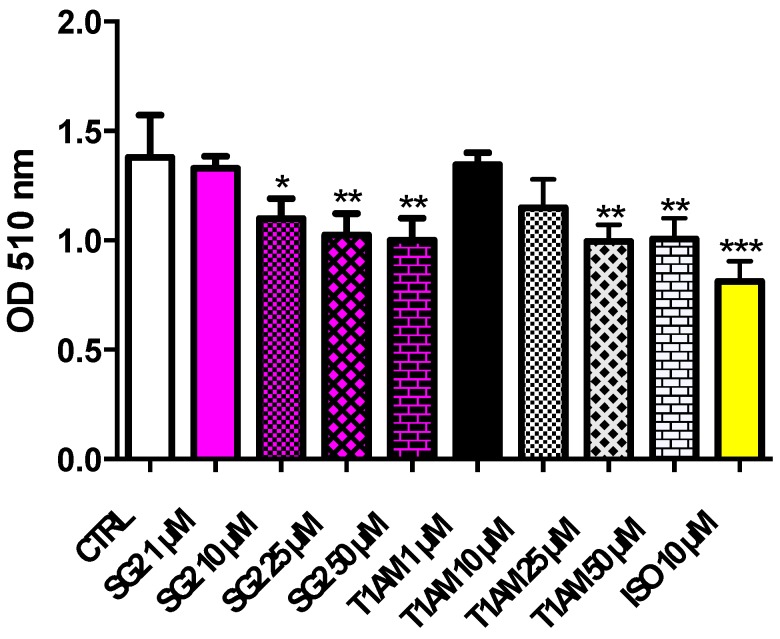
Effects of T1AM and SG-2 on lipid accumulation in fully differentiated 3T3-L1 cells. Mature adipocytes were treated for 24 h with T1AM or SG-2 (1, 10, 25, 50 µM). 10 µM isoproterenol (ISO) was used as positive control. Oil red O stained intercellular oil droplets were eluted with isopropanol and quantified by spectrophotometry analysis at 510 nm. Values represent the mean ± SEM of 4–8 experiments. The groups were compared using the One-Way ANOVA followed by Tukey’s range test. * *p* < 0.05; ** *p* < 0.01; *** *p* < 0.005.

**Figure 6 ijms-20-04054-f006:**
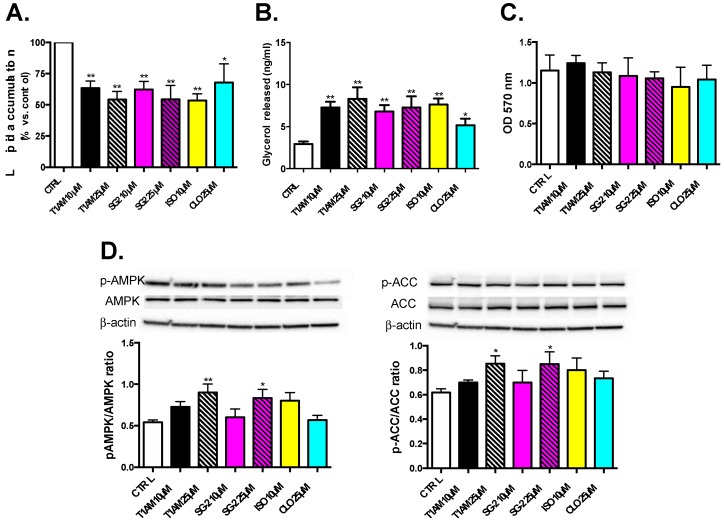
Effects of T1AM and SG-2 on lipids droplets in HepG2 cells. Cells were treated for 24 h with T1AM or SG-2 at two doses (10–25 µM). Cells without treatments labeled as CTRL. A total of 10 µM isoproterenol (ISO) and 25 µM cloroquine (CLO) were used as positive controls. (**A**) Lipid accumulation in HepG2 Cells. Oil Red O stained intercellular oil droplets were eluted with isopropanol and quantified by spectrophotometry analysis at OD = 510 nm. Values represent the mean ± SEM of 4–8 experiments. (**B**) Lipolysis in HepG2 cells. Glycerol released in the culture medium (0.5 mL) of HepG2 cells after 24 h treatment with T1AM or SG-2 at two doses (10 and 25 µM). Values represent the mean ± SEM of 3 to 6 experiments. (**C**) Lack of toxicity in HepG2 cells. Cell viability after 24 h after treatment with T1AM or SG-2. Cell viability was assessed by MTT assay and reported values shown on Y-axis at OD = 570nm. Values represent the mean ± SEM of three independent experiments (**D**) Regulation of metabolism in HepG2 via AMPK and ACC pathways, respectively. Representative immunoblotting images and the quantitative analysis of phosphorylation of AMPK and ACC are shown. The groups were compared using the one-way ANOVA followed by Tukey’s range test. * *p* < 0.05; ** *p* < 0.01.

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
