# Peer review of "Lipolytic Effects of 3-Iodothyronamine (T1AM) and a Novel Thyronamine-Like Analog SG-2 through the AMPK Pathway"

_ijms, 2019, doi:10.3390/ijms20164054_

Round 1

Reviewer 1 Report

Manuscript “Lipolytic effects of 3-iodothyronamine (T1AM) and a novel thyronamine-like analog SG-2 through AMPK pathway” by Rogowski M. et al. deals with an interesting observation where T1AM and its analog SG2 show a great ability to prevent lipid accumulation in adipocytes and hepatocytes. Since lipid accumulation seems to be the main driver of many diseases, including atherosclerosis, glomerular kidney diseases, metabolic diseases, etc., the current study becomes very timely and actual. Rogowski M. et al. also demonstrated partially a possible mechanism of T1AM action, suggesting an active transport of T1AM and its co-localization with mitochondria. This is a well-designed study and a well-written paper. No bias or artifacts should be expected from the methodology used.

However, there are a couple of issues that deserve more attention:

Is decreased lipid accumulation under T1AM and/or SG2 treatment correlated with levels of total and/or esterified cholesterol? In some studies, it has been demonstrated that lipid droplets accumulation observed is due to accumulation of cholesterol esters or total choleterol, but not triglycerides only (please refer to Mitrofanova et al., Kidney International, 2018; Ioannou et al, Hepatol Commun, 2019; etc.). Figure 2: need to label pictures according to the figure legend; please add a bar scale for each sub-figure. Figure 3, B: please add a legend for 100 nM concentration to the graph. Page 5, lane 142: please add data that was not shown for T1AM. It would strengthen your conclusions. Please correct missing Greek symbols across the entire manuscript.

Author Response

Response to the reviewer 1 comments

We thank the reviewer for the constructive comments and criticisms. We have modified the manuscript accordingly.

Reviewer #1: Is decreased lipid accumulation under T1AM and/or SG2 treatment correlated with levels of total and/or esterified cholesterol? In some studies, it has been

demonstrated that lipid droplets accumulation observed is due to accumulation

of cholesterol esters or total choleterol, but not triglycerides only (please refer

to Mitrofanova et al., Kidney International, 2018; Ioannou et al, Hepatol

Commun, 2019; etc.).

Reply: Thank you for this insightful observation. In our study we investigated the effects of SG-2 and T1AM on total lipid accumulation by using the lipid Oil Red O (ORO) staining kit, which is suitable for selective staining and detection of neutral lipids, including triglycerides and cholesterol esters, in cultured cells (Am J Pathol. 1984, 114(2): 201–208). Therefore, from using this assay alone we could not precisely assess whether the decreased lipid accumulation observed after treatment with T1AM and/or SG-2 was correlated with levels of cholesterol. Indeed, cholesterol content could be determined by using a specific staining kit, such as the Amplex Red Cholesterol Assay Kit (Thermo Fisher Scientific, Waltham, MA), as reported by Mitrofanova et al., Kidney International, 2018. In our previous vivo study [1], we measured total serum cholesterol and TG. Our findings showed total serum cholesterol level decreased while total serum TG increased at 25 mg/Kg T1AM, indicating that the pool of cholesterol and TG from major tissues including adipose and liver are changed due to the T1AM treatment. We made changes according to the Reviewer’s suggestion please see Pages 12-13, lines 311-324.

Figure 2: need to label pictures according to the figure legend; please add a bar scale for each sub-figure.

Reply: Done

Figure 3, B: please add a legend for 100 nM concentration to the graph.

Reply: Done

Page 5, line 142: please add data that was not shown for T1AM. It would strengthen your conclusions.

Reply: We did not see any change in color that could visually be recorded. We have modified the text to reflect this observation and removed data not shown.

Please correct missing Greek symbols across the entire manuscript.

Reply: We apologize for symbols not showing up in the PDF version since the word version was OK. We reinserted the symbol throughout the text in the same way to hopefully avoid problems during PDF conversion step.

Reviewer 2 Report

The article of Michael Rogowski et al.  aims at deciphering the cellular mechanism of action of T1AM molecule as  lipolytic agent in different cell lines and particularly in 3T3-L1 mouse adipocytes. Results are of great interest to the field of lipid metabolism and obesity. Experimental design is appropriate and the methodology is adequate. However, several points should be clarified:

Regarding the previous in vivo data and the used dosage of T1AMon which basis the authors have chosen the micromolar concentration to treat 3T3 adipocytes. Figure 1 needs a real legend to the non familiar reader Figure 2 is unclear, authors need to add Green: Lipid stain and Red: FL-104 T1AM images to the presented merged image. Please add labeling A and B on the images. Figures 4: authors should add the image of treated adipocytes at 20 micomolar in correspondence with the histogram (Fig4B). Figure 5: Effects of T1AM and SG-2 on lipid degradation in fully differentiated 3T3-L1 cells. The absence of accumulation doesn’t mean lipid degradation and staining with Oil red O is not a proof of degradation. ATGL, HSL and P-HSL could be used as markers of lypolysis. In addition, authors should evaluate the level of PPARgamma2 during adipocytes differentiation in absence and presence of T1AM, this would answer the effect of T1AM on lipid accumulation.

Author Response

Response to the reviewer’s comments

We thank the reviewer #2 for the constructive comments and criticisms. We have modified the manuscript accordingly.

Reviewer #2: Regarding the previous in vivo data and the used dosage of T1AM on which basis the authors have chosen the micromolar concentration to treat 3T3

adipocytes.

Reply: The reviewer raised an important missing information. The micromolar concentrations used to test the effects of T1AM on lipid accumulation in 3T3-L1 adipocytes and HepG2 cells were chosen to be consistent with previous in vitro experiments carried out by our group to test the metabolic effects of T1AM on hepatocytes ([1],[2]). Our follow up in vivo studies used chronic exposure to T1AM at multiple sub-pharmological doses at 10 and 25 mg/Kg for 5-7 days [2,3]. Based on these studies we used different dosages from nM to mM, in order to explore whether our compounds were able to induce a reduction of lipid accumulation when tested at the same previously studied dosages. We added the information on page 4, lines 104-105. 

Figure 1 needs a real legend to the non familiar reader

Reply: Thank you for this suggestion. A more detailed legend for Figure 1 has been added.

Figure 1. Fluorescently labeled T1AM (FL-T1AM) preparation.Conjugation of Tetramethylrhodamine-6-isothiocyanate (6-TRITC) with T1AM efficiently provides rhodamine labeled T1AM (FL-T1AM).

Figure 2 is unclear, authors need to add Green: Lipid stain and Red: FL-104 T1AM images to the presented merged image. Please add labeling A and B on the

Images.

Reply: Done

Figures 4: authors should add the image of treated adipocytes at 20 micomolar in correspondence with the histogram (Fig4B).

Reply: We did not go beyond 10 micromolar because the effect of SG2 plateaued at the highest 10 mM and there was no further change happened after 5 micromolar concentration. The correction is now made on P7, line 185 to reflect the effect of SG2 vs T1AM.

Figure 5: Effects of T1AM and SG-2 on lipid degradation in fully differentiated 3T3-L1 cells. The absence of accumulation doesn’t mean lipid degradation and staining with Oil red O is not a proof of degradation. ATGL, HSL and P-HSL could be used as markers of lypolysis.

Reply: We agree with this comment. Oil Red O staining exclusively provides a qualitative measure of lipid accumulation in cells. We modified Figure 5’s legend accordingly.

Figure 5: Effects of T1AM and SG-2 on lipid accumulationin fully differentiated 3T3-L1 cells.

Reply: We also modified the title and text on page 9 (lines 167 -172).

In addition, authors should evaluate the level of PPARgamma2 during adipocytes differentiation in absence and presence of T1AM, this would answer the effect of T1AM on lipid accumulation.

Reply: We agree with this comment. PPARg2 would represent the perfect marker to evaluate the effect of T1AM and SG-2 on lipid accumulation during adipocytes differentiation. However, in this study we mainly focused on biochemical studies of two cells lines and explored the impact on T1AM/SG2 uptake into cells. Previously, T1AM gene expression changes were explored to assess lipid accumulation pathways [4]. These pathways will be the subject of our future study to expand our analysis to the transcriptional effects of T1AM analogs on maturing adipocytes.

Ghelardoni, S.; Chiellini, G.; Frascarelli, S.; Saba, A.; Zucchi, R. Uptake and metabolic effects of 3-iodothyronamine in hepatocytes. J Endocrinol 2014, 221, 101-110.

Chiellini, G.; Nesi, G.; Digiacomo, M.; Malvasi, R.; Espinoza, S.; Sabatini, M.; Frascarelli, S.; Laurino, A.; Cichero, E.; Macchia, M., et al.Design, synthesis, and evaluation of thyronamine analogues as novel potent mouse trace amine associated receptor 1 (mtaar1) agonists. J Med Chem 2015, 58, 5096-5107.

Assadi-Porter, F.M.; Reiland, H.; Sabatini, M.; Lorenzini, L.; Carnicelli, V.; Rogowski, M.; Selen Alpergin, E.S.; Tonelli, M.; Ghelardoni, S.; Saba, A., et al.Metabolic reprogramming by 3-iodothyronamine (t1am): A new perspective to reverse obesity through co-regulation of sirtuin 4 and 6 expression. Int J Mol Sci 2018, 19.

Mariotti, V.; Melissari, E.; Iofrida, C.; Righi, M.; Di Russo, M.; Donzelli, R.; Saba, A.; Frascarelli, S.; Chiellini, G.; Zucchi, R., et al.Modulation of gene expression by 3-iodothyronamine: Genetic evidence for a lipolytic pattern. PLoS One 2014, 9, e106923.